# Patient's experience with the Arabin cervical pessary during pregnancy: A questionnaire survey

**Viola Seravalli**ᴼ, **Noemi Strambi**ᴼ, **Alessandra D'Arienzo, Francesco Magni, Ludovico Bernardi, Anna Morucchio, Mariarosaria Di Tommaso** [ID]*

Department of Health Sciences, University of Florence, Florence, Italy

ᴼ These authors contributed equally to this work.
* mariarosaria.ditommaso@unifi.it

**Data Availability Statement:** All relevant data are within the paper and its Supporting Information files.

## Abstract

### Introduction

The cervical pessary is used in women with precocious cervical ripening to prevent preterm birth. Up to now however, there have been no systematic studies on compliance and tolerance, which vary among different study cohorts.

### Material and methods

A questionnaire was administered to 166 women treated with the Arabin cervical pessary in one center. Data were analysed about the patient's experience before insertion (adequacy of information received), during treatment (follow-up, impact on daily life, perceived discomfort, side effects) and at the time of removal (pain, if the patient's expectations had been met regarding the treatment).

### Results

Information received before the insertion of the Arabin cervical pessary was considered adequate in 163/166 (98.2%) women. An increase in vaginal discharge was experienced by 70/166 (42.2%) women. Discomfort or other side effects were reported in 13.8% and 16.3% of cases, respectively. Overall, 77% of women reported an improved quality of life and 94% considered the follow-up during pregnancy adequate. Removal was moderately painful for 58/166 (35%) of women. Patient's expectations regarding the treatment were exceeded in the majority of cases (75.3%). In a final step, we compared our results to previous studies regarding the use of the pessary in singleton and twin pregnancies.

### Conclusion

Although some trials report high rates of non-compliant patients, this could not be confirmed by our study. In contrast, most women reported having a positive experience and that they were motivated to continue the treatment when they were continuously followed by experienced clinicians.

**Funding:** The author(s) received no specific funding for this work.

**Competing interests:** The authors have declared that no competing interests exist.

## Introduction

Globally, preterm birth was reported to have an incidence of 10.6% in 2014 [1], with a wide variation among continents and countries in terms of absolute numbers and rates of preterm births, and in absolute and relative numbers of perinatal and neonatal deaths [2]. Even within Europe, preterm birth rates ranged from 4.9% in Lithuania to 11.2% in Greece in 2015 [3].

Primary prevention of preterm birth would be desirable, but this demands the involvement of publicly funded and supported health policies without immediate benefits, such as smoke-free legislation [3], prevention of teenage pregnancies [4], promotion of healthy diets [5], or possibly the use of medications such as aspirin [6] or omega-3-fatty acids [7–9].

In contrast, secondary prevention of preterm birth describes treatment concepts when the first signs are already recognizable but expected to be reversible. Thereby, a short cervical length (CL) measured by transvaginal sonography is one of the earliest signs, and sonographic assessment of CL is therefore recommended to be applied in high-risk patients or even for screening of the general population [10]. Cervical cerclage, vaginal progesterone and a cervical pessary specially designed to prevent preterm birth are options that are discussed at present for secondary prevention in singleton and twin pregnancies. The question of which method should be chosen does not solely depend on the methods themselves but also whether these pregnant women are followed within dedicated preterm birth clinics by experienced clinicians. Di Renzo et al. in 2017 already recognized the need for clinical training regarding the application of cerclage and cerclage pessaries within the European guidelines for Preterm Birth [11]. However, an adequate practical training is neither described nor audited in many observational or randomized controlled trials.

During the past decade, the cervical pessary has been investigated in different settings in both singleton [12–17] and twin pregnancies [18–24]. It promotes an inclination of the utero-cervical angle as visualized by MRI or clinically [25, 26]. This mechanism is supposed to reduce the pressure on the lower uterine segment at the level of the internal cervical os and the cervix as studied in vivo by a change in maternal position [27, 28], or in vitro by biomechanical engineering [29].

It has been suggested that clinical success also requires experience following a learning curve [30]. Appropriate training of physicians in pessary placement could help to reduce a patient's discomfort during pessary insertion. Up to now, there have been incidental reports on the side effects of the cervical pessary, but women's views and satisfaction rates have not yet been systematically investigated apart from the rates of early removal or discharge within randomized controlled trials (RCTs) [12–16, 18–22].

The discrepant rates of complaints, early removal, and success in preventing preterm birth within both singleton and twin pregnancies motivated us to investigate women's experience with the cervical pessary within our own cohort over 10 years and to compare the results with publications where the consideration of a learning curve was not an issue.

## Material and methods

At Careggi University Hospital in Florence, Italy, a total of 205 women were treated with the Arabin cervical pessary for prevention of preterm birth from June 2010 to June 2020. The treatment was performed by three clinicians who had received proper training. The average treatment per physician was 68.3 insertions in this series, with individual differences. For the two younger physicians, the first 30 applications of the device were supervised by the senior physician who had already had experience with pessary placement. Based on our hospital protocol, the cervical pessary was offered to women with a CL $\leq 25$ mm before 26 weeks of

gestation, with intact membranes. After the insertion of a cervical pessary there was a first follow-up vaginal exam after 48 hours to verify whether the pessary was still surrounding the cervix and had not been displaced, and then follow-up visits were performed by the same clinicians every 2–3 weeks. Only in the case of patients with an extremely short cervical length (<10 mm), an additional transvaginal ultrasound examination after one week was performed to exclude rapid progress of cervical shortening. As an additional treatment, vaginal progesterone was administered to all patients.

Retrospectively, the hospital's electronic database was used to contact the total cohort. 34/205 women (16.6%) could not be contacted because they had changed their contact details and therefore they were not included in the study, while 5/205 (2%) refused to participate. The remaining 166 women were contacted by phone and gave verbal consent to the study. They were administered a questionnaire inquiring about their experience before the insertion (adequacy of the information received), during treatment (follow-up, impact on daily life, perceived discomfort and other side effects) and at the time of removal (presence of pain, degree to which the patient's expectation about the treatment had been met). The questionnaire is available as (S1 File). Numerical rating scales (NRS) from 0 to 10 were used for vaginal discharge and pain. The pain-intensity level was assigned as follows: NRS: 0 = no pain, 1–3 = mild pain, 4–6 = moderate pain, and 7–10 = severe pain [31]. This study was approved by the Institutional Ethics Committee (*Comitato Etico Regionale per la Sperimentazione Clinica della Regione Toscana;* approval number: 18058). Because the study consisted of telephone interviews, the Ethics Committee, complying with our Institution's guidelines, waived the requirement for written consent.

The total cohort and specific subgroups were analysed. In particular, we compared the answers given by Italian women compared to those of foreign nationality, and between women who delivered either before or after 34 weeks, because this more or less defines the success of the treatment, and an earlier delivery might have had an impact on the patient's reported experience or expectations.

Statistical analysis was performed with Graph Pad INSTAT3 software package (San Diego, CA, USA). Continuous variables were expressed as mean and standard deviation; categorical variables were indicated by percentage. We used the chi-square test to compare the answers between the subgroups. A p-value of < 0.05 was considered significant.

## Results

The characteristics of the study cohort of 166 patients that answered the questionnaire are demonstrated in Table 1.

Table 2 summarizes the answers given by the patients. Information received before the insertion of the Arabin pessary was considered adequate in 163/166 cases (98.2%). An increase of vaginal discharge (mean NRS score 5.3), a side effect that is also indicated in the instructions, was experienced by 70/166 (42.2%) women. Discomfort or any other side effects were reported in a minority of cases (13.8% and 16.3%, respectively). Most women (128/166) reported an improved quality of life (77.1%) and even more (94.0%) considered the follow-up, always by the same physician, adequate (Table 2). Removal was moderately painful for 58/166 women (35%), with a mean NRS score of 6.7 ± 2.1. Patient's expectations of treatment outcome were exceeded in the majority of cases (75.3%) and almost all patients (91.6%) reported that they would choose the pessary treatment again or recommend it to a friend in a similar situation (Table 2).

Our study population included 118 Italian and 48 foreign women. There were no significant differences in the answers given between the two subgroups, including side effects (p = 0.49), perceived adequacy of the information (p = 0.56) and of the follow-up received

**Table 1. Mean characteristics of the study group of 166 women treated with the cervical pessary during an observation period of 10 years.**

| Characteristics | Mean ± SD |
|---|---|
| Age (years) | 34.6 ± 5.3 |
| Body mass index (kg/m2) | 22.3 ± 3.6 |
| | **N. (%)** |
| Ethnicity | |
| White | 152 (91.6%) |
| Asian | 9 (5.4%) |
| Black | 5 (3.0%) |
| Nulliparous | 97 (58.4%) |
| Parous | 69 (41.6%) |
| History of preterm birth | 23 (13.9%) |
| Singleton pregnancy | 118 (71%) |
| Multiple pregnancy: | 48 (28.9%) |
| Twin pregnancies | 43 (25.9%) |
| Triplet pregnancies | 5 (3.0%) |
| Spontaneous preterm birth <34 weeks in the index pregnancy: | 46/166 (27.7%) |
| Singleton pregnancies | 26/118 (22.0%) |
| Twin pregnancies | 16/43 (37.2%) |
| Triplet pregnancies | 4/5 (80.0%) |
| pPROM with pessary in situ | 19/166 (11.4%) |
| Singleton pregnancies | 14/118 (11.9%) |
| Twin pregnancies | 3/43 (7.0%) |
| Triplet pregnancies | 2/5 (40.0%) |

(p = 0.33). Among singleton pregnancies, 26/118 (22%) had a spontaneous preterm birth before 34 weeks, while 92/118 (78%) delivered after 34 weeks. 16/43 twin pregnancies (37%) and 4/5 triplet pregnancies (80%) delivered before 34 weeks secondary to spontaneous preterm labor. 3/166 patients (2%) underwent an iatrogenic preterm birth (two for vaginal bleeding in placenta previa, one for HELLP syndrome). One patient (0.6%) required early removal of the pessary due to discomfort. In general, women who delivered later (≥ 34 weeks) more frequently reported an improvement in their daily life, a better than expected experience, and a wish to re-use the device compared to women who delivered before 34 weeks (Table 3).

Finally, we compared our results with respect to the side effects and clinical experience with details reported in studies on the use of the cervical pessary for prevention of preterm birth. We identified one retrospective study and ten RCTs that reported the rates of side effects such as vaginal discharge, discomfort, and pain. The results are demonstrated in Table 4 (studies on singletons) and Table 5 (studies on twins). Although two studies [12, 15] did not report data on training, we have observed a low rate (0–5%) of early removal due to pain or discomfort in studies on singleton pregnancies, including our own, where physicians have received practical training [13, 14, 16] (Table 4). The rate of early removal due to pain or discomfort was higher in the study on twin pregnancies by Norman et al. (11%) [21], where training was provided only by video or on a model. The incidence of vaginal discharge, the most frequent side effect of pessary described by patients (42% in our cohort), is highly variable among the studies (from 14 to 100% in singletons, and from 18 to 100% in twin pregnancies, Tables 4 and 5). Most studies [13–15, 17, 18, 20], including our own, specify that the side effect was an increase in vaginal discharge during treatment compared to before pessary placement (Tables 4 and 5). On the other hand, in three studies [12, 16, 22] reporting a 70–100% vaginal discharge rate

**Table 2. Results of the questionnaire investigating maternal views and experiences before, during and after treatment with cervical pessary (n = 166).**

| Related issues | Possible answers | | |
|---|---|---|---|
| | **n (%)** | | |
| Adequate information before insertion | Yes | No | |
| | 163 (98.2%) | 3 (1.8%) | |
| Information received before insertion regarding possible increase in vaginal discharge | Yes | No | |
| | 128 (77.1%) | 38 (22.9%) | |
| Increased vaginal discharge during the treatment | Yes | No | |
| | 70 (42.2%) | 96 (57.8%) | |
| NRS (Mean ± SD) | 5.3 ± 2.7 | | |
| Any other side effects during the treatment | Yes | No | |
| | 27 (16.3%) | 139 (83.7%) | |
| Change in daily life during the treatment | Yes, positive | Yes, negative | No |
| | 128 (77.1%) | 11 (6.6%) | 27 (16.3%) |
| Discomfort during the treatment | Yes | No | |
| | 23 (13.9%) | 143 (86.1%) | |
| Adequate follow-up | Yes | No | No response |
| | 156 (94.0%) | 4 (2.4%) | 6 (3.6%) |
| Expectations regarding the treatment | Better than I expected | Worse than I expected | As I expected |
| | 125 (75.3%) | 17 (10.2%) | 24 (14.5%) |
| Pain at removal | Yes | No | |
| NRS (Mean ± SD) | 58 (34.9%) | 108 (65.1%) | |
| | 6.7 ± 2.1 | | |
| In a similar situation would you chose the pessary treatment again or recommend it to a friend? | Yes | No | |
| | 152 (91.6%) | 14 (8.4%) | |

NRS, numerical rating scale.

during treatment, the authors do not specify if the discharge increased after the pessary was placed, but they only described a significantly higher rate of this symptom among patients with cervical pessary compared to controls.

**Table 3. Results of the subjective experience stratified for gestational age at delivery (before or after 34 weeks of gestation).**

| | < 34 weeks | ≥ 34 weeks | p-value* |
|---|---|---|---|
| | **50** | **116** | |
| | **n (%)** | **n (%)** | |
| Adequate information before insertion | 47 (94.0%) | 116 (100.0%) | 0.03 |
| Information received before insertion regarding possible vaginal discharge | 37 (74.0%) | 91 (78.4%) | 0.56 |
| Increased vaginal discharge during the treatment | 16 (32.0%) | 54 (46.6%) | 0.09 |
| Any other side effects during the treatment | 10 (20.0%) | 17 (14.7%) | 0.49 |
| Positive changes in daily life during the treatment | 29 (58.0%) | 99 (85.3%) | 0.007 |
| Discomfort during the treatment | 6 (12.0%) | 17 (14.7%) | 0.81 |
| Adequate follow up | 43 (86.0%) | 113 (97.4%) | 0.07 |
| Experience was better than expected | 25 (50.0%) | 100 (86.2%) | < 0.001 |
| Pain at removal | 13 (26.0%) | 45 (38.8%) | 0.16 |
| Would re-use Arabin pessary | 38 (76.0%) | 114 (98.3%) | <0.001 |

* chi square test.

**Table 4. Comparison between studies on pessary use in singleton gestations in terms of side effects, experience and training of clinicians, and clinical results.**

| | Our study | Goya et al. 2012 [12] | Hui et al. 2013 [15] | Saccone et al. 2017 [13] | Nicolaides et al. 2016 [14] | Dugoff et al. 2018 [16] | Ivandic et al. 2020 [17] |
|---|---|---|---|---|---|---|---|
| **Investigated device** | Arabin pessary | Arabin pessary | Arabin pessary | Arabin pessary | Arabin pessary | Bioteque cup pessaries | Arabin pessary |
| **Number of subjects with pessary placed (n)** | 118 | 190 | 53 | 150 | 460 | 60 | 129 |
| **Teaching/ Audit details** | Yes *The treatment was performed by three clinicians who had received practical training in the placement of the device.* | Not specified | Not specified | Yes *The physicians had received practical training in the placement of the device. Pessary insertion training consisted of a didactic session and a hands-on session.* | Yes *The research-team members who inserted the pessaries had received practical training in the placement of the device.* | Yes *In addition to didactic and hands-on training, all staff was required to demonstrate competence in pessary placement on a live model.* | Patients managed by preterm labor team. |
| **Vaginal discharge** | 51.7% (increased during treatment) | 100% | 47.2% (increased during treatment) | 86.7% (increased during treatment) | 10.5% (increased during treatment) | 73.3% | 14.0% (increased during treatment) |
| **Early removal due to discomfort/ pain** | 0.8% | <1% | 0 | 0 | 5.4% | 3% | 5.1% |
| **Discomfort during treatment** | 16.1% | Mean pain score: 4 (scale 0–10) during pessary insertion | 7.5% pressure sensations / 1.9% vaginal pain | 3.3% | 11.4% | 1.7% removal for discomfort during sexual intercourse | 7.0% |
| **Pain during removal (mean score on a 0–10 scale)** | 7.0 | 7 | Not specified | Not specified | Not specified | Not specified | Painful in 28% |
| **Clinical results** | Incidence of PTB<34w: 22% | Cervical pessary associated with significant reduction of PTB < 34w and neonatal composite adverse outcomes in singleton pregnancies with CL ≤25 mm | Cervical pessary not associated with reduction of PTB <34 w in singleton pregnancies with CL<25 mm | Cervical pessary associated with significant reduction of PTB < 34w in asymptomatic singleton pregnancies with CL ≤25 mm | Cervical pessary not associated with reduction of PTB <34 w in singleton pregnancies with CL<25 mm | Cervical pessary not associated with reduction of PTB in singleton pregnancies with short CL<25mm | Incidence of PTB<34w: 28.7% |

PTB, preterm birth. CL, cervical length.

## Discussion

One of the main findings of this study was that apart from vaginal discharge no significant discomfort or side effects were experienced by the vast majority of patients. Moreover, the level of patient satisfaction was high and only one patient required early removal due to discomfort.

In contrast to the PECEP trial [12, 19], in which all the women treated with a cervical pessary had vaginal discharge, this symptom was only present in about 50% of our patients. Other clinical trials report an increase in vaginal discharge in women with pessary treatment compared to women without a pessary in pregnancy, with a highly variable rate reported in the pessary group, depending on the study: from 10.5 in Nicolaides et al. [14] to 73.3% in Dugoff et al. [16] and 86.7% in Saccone et al. [13]. On the other hand, pelvic discomfort is less frequently reported. The lower incidence of vaginal discharge (14%) among women with a

**Table 5. Comparison between studies on pessary use in twin gestations in terms of side effects, experience and training of clinicians, and clinical results.**

| | Our study | Goya et al. 2016 [19] | Dang et al. 2019 [22] | Liem et al. 2013 [18] | Nicolaides et al. 2016 [20] | Norman et al. 2021 [21] |
|---|---|---|---|---|---|---|
| **Investigated device** | Arabin pessary | Arabin pessary | Arabin pessary | Arabin pessary | Arabin pessary | Arabin pessary |
| **Number of subjects with pessary placed (n)** | 48 | 68 | 148 | 401 | 588 | 250 |
| **Teaching/ Audit details** | Yes | Yes | Yes | No | No | Yes (by video) |
| | The treatment was performed by three clinicians who had received practical training in the placement of the device. | The central team in turn instructed the other centers in the use of the pessary | Well-trained staff involved in pessary treatment | No specific training was provided | Many research team doctors were involved in the insertion of the pessary and they did not receive supervised training in doing so | Inserting obstetricians watched a training video on pessary insertion, were provided with written guidance on pessary management, and (at their discretion) practiced pessary insertion on a model prior to first insertion |
| **Vaginal discharge** | 18.7% (increased during treatment) | 100% | 70% | 26% (increased during treatment) | 42.1% (increased during treatment) | Not specified |
| **Early removal due to discomfort/ pain** | 0 | Not specified | Not specified | 5.7% | 5% | 11.3% |
| **Discomfort during treatment** | 8.3% | Mean pain score: 4 (scale 0–10) during pessary insertion | Discomfort (17%) Pain 4% | 4% | 5.8% | 11.3% (32.5% discomfort or pain during insertion) |
| **Pain during removal (mean score on a 0–10 scale)** | 5.7 | 7 | Not specified | Not specified | Not specified | Uncomfortable in 41.3% |
| **Clinical results** | Incidence of PTB<34w: 37% | Cervical pessary associated with significant reduction of PTB < 34w in twin pregnancies with short CL ≤25 mm | Cervical pessary associated with reduction of PTB < 34w and improved composite poor perinatal outcome in twin pregnancies with CL <28 mm | Cervical pessary associated with reduction of composite poor perinatal outcome in pregnancies with a CL< 38 mm between 16 and 20 w | Cervical pessary not associated with reduction of PTB <34w in unselected twin pregnancies. | Cervical pessary not associated with reduction of PTB <34w nor composite adverse neonatal outcome in twin pregnancies with CL ≤35 mm |

PTB, preterm birth. CL, cervical length.

pessary reported in the study by Ivandic et al. [17] compared to most RCTs may be explained by the retrospective nature of the study (taking part in a clinical trial may undermine patient confidence in the treatment, resulting in more dissatisfaction or anxiety regarding the treatment) and by the fact that they reported the occurrence of "significant" vaginal discharge.

Since increased vaginal discharge is the most common side effect of pessary treatment in most studies, the patient should be advised of this before the pessary is positioned. As our study shows, 77.1% of women reported that they had received adequate, comprehensive information about the possibility of vaginal discharge. Communication is an essential part of the treatment, as accurate counselling can be helpful in increasing patient's compliance and satisfaction regarding the treatment.

In order to prevent the accumulation of vaginal fluids, we encourage the use of Arabin cervical pessaries which are characterized by perforations in the silicone ring that favour the release of vaginal discharge [23]. It is important to keep in mind that vaginal discharge due to

the Arabin pessary is not a sign of infection and should not be treated with antibiotics, unless there is evidence of infection based on positive vaginal cultures.

Encouraging data in relation to the impact of the treatment on a patient's daily life has emerged from our study. The fact that the majority of patients have reported positive changes in their lives during treatment may be in part related to the decreased concern about the risk of preterm birth. In addition, the presence of the pessary supporting the cervix may give relief of pressure sensations while walking or standing in some patients [26].

Patients of foreign nationality did not report a different experience with the treatment, or a different perception of the assistance and information received compared to Italian patients, while women who delivered before 34 weeks reported adequacy of the information received, positive changes in their daily life, a sense of satisfaction and consideration of a possible reuse of the device with a lower frequency compared to women who delivered after 34 weeks. This could be explained by the fact that childbirth occurred at low gestational ages and, consequently, in some women a negative experience related to the prematurity of their child may also have had an impact on the reported experience with the pessary.

This is the first study which has focused on the maternal experience regarding pessary treatment. A strength of this study is that all the women included in the study were assisted by operators who had been trained in proper pessary placement. This study also has limitations, which include the retrospective design, the absence of a control group and the fact that a pilot test to validate the questionnaire was not conducted. Moreover, we acknowledge that our study may be limited by recall bias, as the participants were asked to answer questions about past experiences and outcomes. Therefore, the accuracy of the answers provided may be less reliable when compared to data collected prospectively during research trials. To reduce recall bias, we used a carefully compiled questionnaire, with very specific questions, in order to maximize accuracy and completeness.

## Conclusions

Although some randomized trials report high rates of non-compliant patients, this could not be confirmed by our data. In contrast, most women treated with the Arabin pessary for prevention of preterm birth reported a positive experience, and the main side effect was an increase in vaginal discharge. Women were motivated to continue with the treatment when they were continuously followed by experienced clinicians.

## Supporting information

**S1 File. Questionnaire administered to the study population.**
(DOCX)

**S1 Dataset.**
(SAV)

## Author Contributions

**Conceptualization:** Viola Seravalli, Noemi Strambi, Mariarosaria Di Tommaso.

**Data curation:** Viola Seravalli, Noemi Strambi, Alessandra D'Arienzo, Francesco Magni, Ludovico Bernardi, Anna Morucchio.

**Formal analysis:** Viola Seravalli, Alessandra D'Arienzo, Francesco Magni.

**Investigation:** Alessandra D'Arienzo, Ludovico Bernardi, Anna Morucchio.

**Methodology:** Viola Seravalli, Noemi Strambi, Ludovico Bernardi, Anna Morucchio.

**Software:** Alessandra D'Arienzo.

**Supervision:** Viola Seravalli, Mariarosaria Di Tommaso.

**Validation:** Viola Seravalli, Noemi Strambi, Francesco Magni, Mariarosaria Di Tommaso.

**Writing – original draft:** Viola Seravalli, Noemi Strambi.

**Writing – review & editing:** Alessandra D'Arienzo, Francesco Magni, Ludovico Bernardi, Anna Morucchio, Mariarosaria Di Tommaso.

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
