## [Decision Letter · Decision Letter 0]

7 Oct 2021

PONE-D-21-15296Patient’s experience with Arabin cervical pessary during pregnancy: a questionnaire surveyPLOS ONE

Dear Dr. Di Tommaso,

Thank you for submitting your manuscript to PLOS ONE. After careful consideration, we feel that it has merit but does not fully meet PLOS ONE’s publication criteria as it currently stands. Therefore, we invite you to submit a revised version of the manuscript that addresses the points raised during the review process.

We look forward to receiving your revised manuscript.

Kind regards,

Antonio Simone Laganà, M.D., Ph.D.

Academic Editor

PLOS ONE

Journal Requirements:

2. Please amend your current ethics statement to address the following concerns: Please explain why written consent was not obtained, how you recorded/documented participant consent, and if the ethics committees/IRBs approved this consent procedure."

3. Please state whether you validated the questionnaire prior to testing on study participants. Please provide details regarding the validation group within the methods section.

Additional Editor Comments (if provided):

The reviewers have expressed positive comments regarding your article, raising only few concerns. Considering this point, I invite authors to perform the required minor revisions.

Reviewers' comments:

Reviewer's Responses to Questions

**Comments to the Author**

1. Is the manuscript technically sound, and do the data support the conclusions?

Reviewer #1: Partly

Reviewer #2: Yes

Reviewer #3: Yes

Reviewer #4: Yes

2. Has the statistical analysis been performed appropriately and rigorously? 

Reviewer #1: No

Reviewer #2: Yes

Reviewer #3: Yes

Reviewer #4: Yes

3. Have the authors made all data underlying the findings in their manuscript fully available?

Reviewer #1: Yes

Reviewer #2: Yes

Reviewer #3: Yes

Reviewer #4: Yes

4. Is the manuscript presented in an intelligible fashion and written in standard English?

Reviewer #1: Yes

Reviewer #2: Yes

Reviewer #3: Yes

Reviewer #4: Yes

5. Review Comments to the Author

Reviewer #1: Overall you have provided a reasonable summary of patient experience of treatment with Arabin pessary over a 10 year period from a single centre. However the authors make the assumption that clinical experience of healthcare specialists explains the difference in previously reported prospectively collected RCT results. The manuscript does not provide sufficient evidence to support this statement. Please revise.

To strengthen this manuscript more information about the clinical protocol, clinical outcomes and alternative therapies would be useful.

Minor comments:

Line 44: You have included ‘policies to reduce physical stress’ from your list of evidence-based supported healthcare programmes and used a reference from 1985 in support. There has since been plenty of evidence that physical stress / exercise in pregnancy reduces rates of preterm birth. Please justify/clarify this statement, provide additional evidence or consider removing ‘policies to reduce physical stress’ from your list of evidence-based supported healthcare programmes.

Line 63: Formatting of references changes for 27, please amend.

Line 55 suggest use of word ‘recognised’ instead of ‘demanded’. i.e. ‘Di Renzo et al. in 2017 recognised the need for clinical training for the insertion of cerclage and cerclage pessaries’

Line 77: please use a unit of measurement for 68.3 (? Insertions)

Line 78: what do you mean by second control after 48 hours? Please clarify.

Line 79: please provide a definition for extremely short cervical length? (i.e. <15mm, <5mm)

Line 80: When was an additional TVU performed – immediately, one week later?

Line 82: How did you know that 34 women had changed address? Please clarify how you contacted them and who contacted them before the questionnaire was sent out (i.e. their clinical team or a research team).

In table 4 please justify why you have not included Hui 2003, Daskalakis 2013 and Cabrera Garcia 2015.

In table 4 please explain what is meant by “extensive training” – column one.

Table 4 is difficult to read and not all the details are necessary – please could you make more concise – particularly the section on clinical results. Consider additional rows to compare information on cervical length and gestation for inclusion/insertion then include rates of sPTB.

Major comments

Methods: Please make the management protocol of these women clear. How were these women screened for treatment? Did they have follow up with the same individuals? How many times were they seen in their pregnancy? Was TVU was performed after the pessary was placed? Did the management protocol change over the 10 year period, and if so, how?

You make the point in your introduction that the clinical success with Arabin pessary follows a learning curve – which ‘has been completely neglected’ and experienced clinicians improve the satisfaction rate. This point is also reiterated in your abstract results “In a final step, we compared our results to previous studies suggesting that clinical experience of the health care specialists in charge may explain why the discrepancy of results”. How much experience prior to 2010 did your clinicians have with insertion of pessaries? If you provided a questionnaire to all the women included in this case series you may expect to see an improvement with satisfaction over time from 2010 to 2020, otherwise you fail to provide evidence for the need for experienced clinicians, unless they were highly experienced before 2010, however Arabin pessaries were not widely used. Please clarify how these three clinicians obtained their experience prior to 2010, and how is this different to the RCT training?

It would be useful to include data showing the number of pessaries inserted per year and relative satisfaction rates. i.e. did clinician confidence improve with use over time and more pessaries offered – or was confidence always high and treatment rates remained equal each year?

In table 4 – at least 6 RCTs appear to have provided adequate training in the placement of the pessary. You have not commented on if these 6 RCTs report significantly different results of patient experience / Arabin failure to the RCTs that did not provide treatment or have not reported on evidence of training. At present I do not feel that the data you have provided supports this assertion. Please provide adequate evidence and consider alternative explanations for the variation in reported results.

You summarise the data on RCT in Arabin pessaries but it is unhelpful to put clinical results of singleton and twin studies in the same table. Please separate these tables into a singleton and twin tables to make clinical comparison more useful for the reader.

Additionally please specifically address in your study weaknesses the likely recall bias, if you are asking women to remember their experiences of quantities of vaginal discharge between 6 months and 10 years after treatment. How reliable do you think these results are when compared to data that was collected prospectively during research trials?

There is huge variation in reports of vaginal discharge? How are RCTs collecting information and reporting on vaginal discharge? Were they asking women if there was an increase before and after treatment (no matter how small) or were they asking if it was troublesome vaginal discharge (i.e. causing them to wear pads etc.). Please address possible reasons for this difference in your discussion.

Table 2 – the answers here are reporting percentages of yes and no. Please justify what the addition of a statistical p-value adds to this data and consider removal of this column. i.e. there is a statistical significant difference in the number of women who received information before insertion regarding possible discharge, but 22.9% of women did not receive information. Do you think clinically this is acceptable when 42.2% of women report vaginal discharge during treatment?

Taking part in a research trial may undermine patient confidence in the treatment, resulting in more dissatisfaction or anxiety with the treatment. This may therefore not be the be best comparison for your retrospective study – consider comparing your results to other retrospective studies of Arabin pessary experience from a single centre in your discussion to strengthen your findings and put them in context (e.g. Jelena Ivandic, Angharad Care, Laura Goodfellow, Borna Poljak, Andrew Sharp, Devender Roberts & Zarko Alfirevic (2020) Cervical pessary for short cervix in high risk pregnant women: 5 years experience in a single centre, The Journal of Maternal-Fetal & Neonatal Medicine, 33:8, 1370-1376, DOI: 10.1080/14767058.2018.1519018).

This study design does suffer with the innate weakness of recall bias. I feel it would improve the manuscript to include more data about the overall clinical experience with the pessary. i.e. the reasons for intervention (? All short cervix <25mm), length of cervix on insertion, gestation at insertion, if additional adjunctive treatment was used (i.e. progesterone), how many patients required removal of pessary, how many required alternative treatments (i.e. rescue cerclage) and rate of PPROM with pessary in situ.

Reviewer #2: The manuscript entitled "Patient’s experience with Arabin cervical pessary during pregnancy: a questionnaire survey" analyzes the compliance and tolerance of pregnant women wearing Arabin cervical pessary. The authors concluded that the results cannot fully confirm a positive attitude of women towards this pessary, but more evidence is needed to investigate this topic.

The manuscript presents an interesting topic that falls within the scope of the journal “Plos One”. The Methodology is well written and the tables are detailed. The results are well represented, and the discussion is fine as the limitations of the study are explained.

I suggest the authors to discuss, at least briefly, about the universal screening for preterm delivery by cervical length measurement during the mid trimester scan (PMID: 29536576).

Reviewer #3: The manuscript entitled "Patient’s experience with Arabin cervical pessary during pregnancy: a questionnaire survey" analyzes the compliance and tolerance of pregnant women wearing Arabin cervical pessary. The authors concluded that the results cannot fully confirm a positive attitude of women towards this pessary, but more evidence is needed to investigate this topic.

The manuscript presents an interesting topic that falls within the scope of the journal “Plos One”. The Methodology is well written and the tables are detailed. The results are well represented, and the discussion is fine as the limitations of the study are explained.

I suggest the authors to discuss, at least briefly, about the universal screening for preterm delivery by cervical length measurement during the mid trimester scan (PMID: 29536576).

Reviewer #4: I read with great interest the manuscript titled "Patient's experience with Arabin cervical pessary during pregnancy: a questionnaire survey" (PONE-D-21-15296).

The topic of this manuscript falls within the scope of Plos One.

I was particularly pleased to review this paper. In my honest opinion, the topic is interesting enough to attract the readers' attention. Moreover, the methodology is accurate, and the data analysis supports conclusions. Nevertheless, the authors should clarify some points and improve the discussion.

In general, the manuscript may benefit from several minor revisions, as suggested below:

- All the text needs a minor language revision to correct some typos and grammatical errors.

- I would suggest reporting indications for Arabin pessary use in the methods section, such as inclusion and exclusion criteria of patients who were eligible for Arabin pessary use.

- I would suggest providing some details about how the authors identified the articles reported in table 4.

- Lines 162-163. I would suggest clarifying this sentence better. It is unclear.

- Strengths and limitations should be better discussed. The "experience" of the operator is not a standardized definition and is not comparable with other studies. Therefore, statements supporting the role of operator experience cannot be supported by study results. Regarding study limitations, the main limitation is not reported and discussed by the authors: the recall bias. Based on study methods, study outcomes were assessed by interview and getting in touch with patients up to 20 years after pessary use, providing concerns regarding the risk of recall bias.

6. PLOS authors have the option to publish the peer review history of their article (what does this mean?). If published, this will include your full peer review and any attached files.

Reviewer #1: **Yes: **Dr Angharad Care

Reviewer #2: No

Reviewer #3: No

Reviewer #4: No

---

## [Author Response · Author response to Decision Letter 0]

23 Nov 2021

Reviewer #1: Overall you have provided a reasonable summary of patient experience of treatment with Arabin pessary over a 10 year period from a single centre. However the authors make the assumption that clinical experience of healthcare specialists explains the difference in previously reported prospectively collected RCT results. The manuscript does not provide sufficient evidence to support this statement. Please revise.

R. We thank the reviewer for the remark. We have removed the statement “..clinical experience of the health care specialists in charge may explain why the discrepancy of results” from the abstract and we have revised our statement in the introduction (line 65 of the clean version, now as follows: It has been suggested that clinical success also requires experience following a learning curve , ref. to Franca et al 2020 ). The large discrepancies in outcomes between studies can be explained by several reasons, including distinct selection criteria (e.g. different cut-off values of the cervical length in different studies), ethnic and cultural particularities. We believe that one point that is often neglected is the importance of training on how to handle the cervical pessary and that a proper training in pessary placement may facilitate less maternal discomfort during the insertion of the device by trained physicians. However, we agree with the reviewer that statements supporting the role of operator experience cannot be supported by our study results. Therefore, we removed the sentence: “There seems to be a negative association of increased early removal with the experience of the clinicians involved in the treatment and the practical training received”, and we only report instead that low rates (0-5%) of early removal due to pain or discomfort are observed in studies on singleton pregnancies, including ours, where physicians received practical training. 

To strengthen this manuscript more information about the clinical protocol, clinical outcomes and alternative therapies would be useful. R. We agree with the reviewer, and we have now included in the Methods more details about the clinical protocol for pessary use at our Center and about additional treatment (progesterone). Data on outcome are reported in Table 1.

Minor comments:

Line 44: You have included ‘policies to reduce physical stress’ from your list of evidence-based supported healthcare programmes and used a reference from 1985 in support. There has since been plenty of evidence that physical stress / exercise in pregnancy reduces rates of preterm birth. Please justify/clarify this statement, provide additional evidence or consider removing ‘policies to reduce physical stress’ from your list of evidence-based supported healthcare programmes. R. We agree with the reviewer that physical exercise in pregnancy is recommended, even in women at risk for preterm birth, as it has shown to reduce the risk of PTD. The sentence was meant to refer to possible effects of mental and physical stress related to excessive workload in pregnancy. However, we have now removed the reference and the sentence because it was misleading.

Line 63: Formatting of references changes for 27, please amend. R. The formatting has been corrected.

Line 55 suggest use of word ‘recognised’ instead of ‘demanded’. i.e. ‘Di Renzo et al. in 2017 recognised the need for clinical training for the insertion of cerclage and cerclage pessaries’. R. The verb has been changed as suggested

Line 77: please use a unit of measurement for 68.3 (? Insertions) R. The unit has been specified

Line 78: what do you mean by second control after 48 hours? Please clarify.R. We meant a follow-up vaginal exam to verify the correct position of the pessary. We have now explained it more clearly in the sentence.

Line 79: please provide a definition for extremely short cervical length? (i.e. <15mm, <5mm) R. A definition has been provided (< 10 mm)

Line 80: When was an additional TVU performed – immediately, one week later? R. It is usually performed one week later in case of CL<10 mm, based on our clinical protocol. This is now specified in the methods.

Line 82: How did you know that 34 women had changed address? Please clarify how you contacted them and who contacted them before the questionnaire was sent out (i.e. their clinical team or a research team). R. We tried to call those women but the phone number we had was no longer in use, therefore we assumed that they had changed phone number and they were not included in the study. 

In table 4 please justify why you have not included Hui 2003, Daskalakis 2013 and Cabrera Garcia 2015.R We thank the reviewer for the remark. We have now included the work by Hui 2013 in Table 4 (singletons). We did not include the work by Daskalakis 2013 because it is a review and we meant only to list the clinical studies in table 4. We have found a study protocol published by Cabrera Garcia in 2015 for a RCT (PESAPRO trial). However, we were not able to find the results of such trial.

In table 4 please explain what is meant by “extensive training” – column one. R. We have changed the term “extensive traning” with “practical training in the placement of the device”, by which we mean that the younger physicians were supervised by the senior physician (who already had experience with the pessary) for the first 30 applications of the pessary (with the senior physician checking the correct position of the pessary), before starting to place pessaries without supervision. Details of such training are now also included in the Methods (line 79-81 of the clean version).

Table 4 is difficult to read and not all the details are necessary – please could you make more concise – particularly the section on clinical results. Consider additional rows to compare information on cervical length and gestation for inclusion/insertion then include rates of sPTB. R. We thank the reviewer for the suggestions. We have modified the section on clinical results of table 4, which is now more concise. The results are now also more easily comparable between studies, and the different cut offs of CL are more clearly reported. We have also divided the original table in two tables separating studies on singletons (table 4) and twins (Table 5) to make clinical comparison more useful for the reader, as suggested by the reviewer (see major comments).

Major comments

Methods: Please make the management protocol of these women clear. How were these women screened for treatment? Did they have follow up with the same individuals? How many times were they seen in their pregnancy? Was TVU was performed after the pessary was placed? Did the management protocol change over the 10 year period, and if so, how? R. Pessary treatment was offered to women with a CL ≤25 mm before 26 weeks of gestation, who had intact membranes. After insertion, they were followed up by the same clinicians with obstetric visits every 2-3 weeks. TVU was performed after 1 week of pessary placement in case of CL<10 mm (see Methods). This protocol has been remained the same over the 10 year period. All the details about the protocol are now included in the methods.

You make the point in your introduction that the clinical success with Arabin pessary follows a learning curve – which ‘has been completely neglected’ and experienced clinicians improve the satisfaction rate. This point is also reiterated in your abstract results “In a final step, we compared our results to previous studies suggesting that clinical experience of the health care specialists in charge may explain why the discrepancy of results”. How much experience prior to 2010 did your clinicians have with insertion of pessaries? If you provided a questionnaire to all the women included in this case series you may expect to see an improvement with satisfaction over time from 2010 to 2020, otherwise you fail to provide evidence for the need for experienced clinicians, unless they were highly experienced before 2010, however Arabin pessaries were not widely used. Please clarify how these three clinicians obtained their experience prior to 2010, and how is this different to the RCT training? R. We thank the reviewer for the remarks. We have removed the statement “..the clinical experience of the health care specialists in charge may explain why the discrepancy of results” from the abstract, as we agree with the reviewer that our results do not provide sufficient evidence to support this statement. We also revised our statement in the introduction (line 65 of the clean version), reporting that “It has been suggested” that clinical success also requires experience following a learning curve (ref. to the study by Franca et al.2021).

We have now included more details about our experience (line 79-81 of the clean version): “for the two younger physicians, the first 30 applications of the device were supervised by the senior physician who already had experience with pessary placement.” The senior obstetrician had started placing pessary since 2008 (following the results of Arabin et al. Is treatment with vaginal pessaries an option in patients with a sonographically detected short cervix? J Perinat Med. 2003;31(2):122-33. which suggested that its use in patients at risk for PTB was cost-effective). We were unable to observe an improvement in patient’s satisfaction over time from 2010 to 2020, given the small number of patients per year in our study (e.j. only a total of 15 patients were included in the first three years 2010-2012)

It would be useful to include data showing the number of pessaries inserted per year and relative satisfaction rates. i.e. did clinician confidence improve with use over time and more pessaries offered – or was confidence always high and treatment rates remained equal each year? R. We were unable to observe an improvement with satisfaction over time from 2010 to 2020, given the small number of patients per year in our study (e.j. 15 patients were included in the first three years 2010-2012). The number of patients treated increased over the years, possibly suggesting: i) increased clinician’s confidence, and ii) increased evidence supporting pessary use coming from studies performed over the years.

In table 4 – at least 6 RCTs appear to have provided adequate training in the placement of the pessary. You have not commented on if these 6 RCTs report significantly different results of patient experience / Arabin failure to the RCTs that did not provide treatment or have not reported on evidence of training. At present I do not feel that the data you have provided supports this assertion. Please provide adequate evidence and consider alternative explanations for the variation in reported results. R. We thank the reviewer for the remark. We agree that we are not able to demonstrate a negative association of increased early removal with the experience of the clinicians (this statement was removed from the Results), also because some of the trials do not specify the rate of early removal, nor give any information on patient’s satisfaction rate. Therefore, we only “observed low rates (0-5%) of early removal due to pain or discomfort in studies on singleton pregnancies, including ours, where physicians received practical training”

You summarise the data on RCT in Arabin pessaries but it is unhelpful to put clinical results of singleton and twin studies in the same table. Please separate these tables into a singleton and twin tables to make clinical comparison more useful for the reader. R. We have now created two different tables for studies on singletons (table 4) and twins (table 5)

Additionally please specifically address in your study weaknesses the likely recall bias, if you are asking women to remember their experiences of quantities of vaginal discharge between 6 months and 10 years after treatment. How reliable do you think these results are when compared to data that was collected prospectively during research trials?

R. We agree with the reviewer that recall bias is an important limitation of our study, and we have addressed it in the discussion.

There is huge variation in reports of vaginal discharge? How are RCTs collecting information and reporting on vaginal discharge? Were they asking women if there was an increase before and after treatment (no matter how small) or were they asking if it was troublesome vaginal discharge (i.e. causing them to wear pads etc.). Please address possible reasons for this difference in your discussion: R. Unfortunately, details on how information about vaginal discharge was collected are quite scarce in the RCTs analyzed. The most detailed description of how women were counselled about the possibility of vaginal discharge was reported in the retrospective study by Ivandic et al (2018), who reported a 14% incidence of “significant vaginal discharge” with pessary. In our questionnaire, we asked the patients if they experienced an increase of vaginal discharge with the treatment (the question is now more clearly reported in table 2). We reviewed all the other studies: most of them (Saccone et al, Hui et al, Ivandic et al, Nicolaides et al, Liem et al.) specify that the side effect was “increased vaginal discharge” with treatment compared to before treatment (this is now specified in table 4 and 5). On the other hand, three studies (ref Goya et al ,Dugoff et al, Dang et al) ,which report a 70-100% vaginal discharge rate during treatment, do not specify if there was an increase after treatment, but only compared the rate of vaginal discharge between women with pessary and controls, reporting a statistically significant difference. We have included these comments in the Results. 

Table 2 – the answers here are reporting percentages of yes and no. Please justify what the addition of a statistical p-value adds to this data and consider removal of this column. i.e. there is a statistical significant difference in the number of women who received information before insertion regarding possible discharge, but 22.9% of women did not receive information. Do you think clinically this is acceptable when 42.2% of women report vaginal discharge during treatment? R. We have removed the column with the p-value, as we agree that it does not add any relevant information. After reviewing the results of the questionnaire, the fact that 23% of our patients did not recall having received information about such a common side effect forced us to review the way we provide the counselling and to improve it.

Taking part in a research trial may undermine patient confidence in the treatment, resulting in more dissatisfaction or anxiety with the treatment. This may therefore not be the be best comparison for your retrospective study – consider comparing your results to other retrospective studies of Arabin pessary experience from a single centre in your discussion to strengthen your findings and put them in context (e.g. Jelena Ivandic, Angharad Care, Laura Goodfellow, Borna Poljak, Andrew Sharp, Devender Roberts & Zarko Alfirevic (2020) Cervical pessary for short cervix in high risk pregnant women: 5 years experience in a single centre, The Journal of Maternal-Fetal & Neonatal Medicine, 33:8, 1370-1376, DOI: 10.1080/14767058.2018.1519018). R We have reviewed the article by Ivandic et al, and added the reference to that paper in our manuscript. The comparison of their results with those of our study is also now included in the table. As stated above, the most detailed description of how women were counselled about vaginal discharge was reported in the retrospective study by Ivandic et al (2018), who reported a 14% incidence of “significant vaginal discharge” with pessary. This is now commented in the discussion.

This study design does suffer with the innate weakness of recall bias. I feel it would improve the manuscript to include more data about the overall clinical experience with the pessary. i.e. the reasons for intervention (? All short cervix <25mm), length of cervix on insertion, gestation at insertion, if additional adjunctive treatment was used (i.e. progesterone), how many patients required removal of pessary, how many required alternative treatments (i.e. rescue cerclage) and rate of PPROM with pessary in situ. R. We have now included the recall bias among the limitations of our study. In addition, based on the reviewer’s suggestion, we have specified the indications for the interventions (cut-off of CL and Gestational age: see details about clinical protocol for pessary treatment in the Methods section). As additional treatment, vaginal progesterone was administered to all singletons (see Methods). The overall rate of pPROM with pessary in situ was 11.4% (19/166) (now reported in table 1). The rate of removal is reported in table 4/5. 

Reviewer #2: The manuscript entitled "Patient’s experience with Arabin cervical pessary during pregnancy: a questionnaire survey" analyzes the compliance and tolerance of pregnant women wearing Arabin cervical pessary. The authors concluded that the results cannot fully confirm a positive attitude of women towards this pessary, but more evidence is needed to investigate this topic.

The manuscript presents an interesting topic that falls within the scope of the journal “Plos One”. The Methodology is well written and the tables are detailed. The results are well represented, and the discussion is fine as the limitations of the study are explained.

I suggest the authors to discuss, at least briefly, about the universal screening for preterm delivery by cervical length measurement during the mid trimester scan (PMID: 29536576). R. We thank the reviewer for the suggestion. In the introduction section, we have included a comment on the recommendation to universal cervical screening (line 52 of the clean version), and we have now added the reference to the work by Souka et al (2019), as suggested.

Reviewer #3: The manuscript entitled "Patient’s experience with Arabin cervical pessary during pregnancy: a questionnaire survey" analyzes the compliance and tolerance of pregnant women wearing Arabin cervical pessary. The authors concluded that the results cannot fully confirm a positive attitude of women towards this pessary, but more evidence is needed to investigate this topic.

The manuscript presents an interesting topic that falls within the scope of the journal “Plos One”. The Methodology is well written and the tables are detailed. The results are well represented, and the discussion is fine as the limitations of the study are explained.

I suggest the authors to discuss, at least briefly, about the universal screening for preterm delivery by cervical length measurement during the mid trimester scan (PMID: 29536576). ). R. We thank the reviewer for the suggestion. In the introduction section, we have included a comment on the recommendation to universal cervical screening (line 52 of the clean version), and we have now added the reference to the work by Souka et al (2019), as suggested.

Reviewer #4: I read with great interest the manuscript titled "Patient's experience with Arabin cervical pessary during pregnancy: a questionnaire survey" (PONE-D-21-15296).

The topic of this manuscript falls within the scope of Plos One.

I was particularly pleased to review this paper. In my honest opinion, the topic is interesting enough to attract the readers' attention. Moreover, the methodology is accurate, and the data analysis supports conclusions. Nevertheless, the authors should clarify some points and improve the discussion.

In general, the manuscript may benefit from several minor revisions, as suggested below:

- All the text needs a minor language revision to correct some typos and grammatical errors. R. The manuscript has now been reviewed by a native English speaker to correct any grammatical or typos errors.

- I would suggest reporting indications for Arabin pessary use in the methods section, such as inclusion and exclusion criteria of patients who were eligible for Arabin pessary use. R. We have now specified in the Methods section that cervical pessary was offered to women with a CL ≤25 mm before 26 weeks of gestation, who had intact membranes.

- I would suggest providing some details about how the authors identified the articles reported in table 4. R. We identified the studies on pessary that reported the rates of side effects such as vaginal discharge, discomfort, and pain. Therefore, we included all randomized clinical trials on pessary use, now reported in Table 4 (singletons) and Table 5 (twins) (as suggested by Reviewer #1). In addition, we included a retrospective study by Ivandic et al (2018), as requested by Reviewer #1, that describe the 5-years experience with pessary in a single center, and reported the rates of side effects. Details are now provided in the Results section.

- Lines 162-163. I would suggest clarifying this sentence better. It is unclear. R. We have now replaced the sentence with the following: “Most importantly, it should be remembered that vaginal discharge due to Arabin pessary is not a sign of infection and should not be treated with antibiotics.” 

- Strengths and limitations should be better discussed. The "experience" of the operator is not a standardized definition and is not comparable with other studies. Therefore, statements supporting the role of operator experience cannot be supported by study results. Regarding study limitations, the main limitation is not reported and discussed by the authors: the recall bias. Based on study methods, study outcomes were assessed by interview and getting in touch with patients up to 20 years after pessary use, providing concerns regarding the risk of recall bias. 

R 1. We thank the reviewer for the remark. We have removed the statement “clinical experience of the health care specialists in charge may explain why the discrepancy of results” from the abstract and we have revised our statement in the introduction (line 65 of the clean version, now as follows: It has been suggested that clinical success also requires experience following a learning curve, , ref. to Franca et al 2021). The large discrepancies in outcomes between studies can be explained by several reasons, including distinct selection criteria (e.g. different cut-off values of the cervical length in different studies), ethnic and cultural particularities. We also believe that one point that is often neglected is the importance of training on how to handle the cervical pessary. However, we agree with the reviewer that statements supporting the role of operator experience cannot be supported by study results. Therefore, we removed the sentence: “There seems to be a negative association of increased early removal with the experience of the clinicians involved in the treatment and the practical training received”, and we only report instead that low rates (0-5%) of early removal due to pain or discomfort are observed in studies on singleton pregnancies, including ours, where physicians received practical training. 

2. We agree with the reviewer that recall bias is an important limitation of our study, and we have now addressed it in the discussion.

---

## [Decision Letter · Decision Letter 1]

13 Dec 2021

Patient’s experience with the Arabin cervical pessary during pregnancy: a questionnaire survey

PONE-D-21-15296R1

Dear Dr. Di Tommaso,

We’re pleased to inform you that your manuscript has been judged scientifically suitable for publication and will be formally accepted for publication once it meets all outstanding technical requirements.

Kind regards,

Antonio Simone Laganà, M.D., Ph.D.

Academic Editor

PLOS ONE

Additional Editor Comments (optional):

I carefully evaluated the revised version of this manuscript.

Authors have performed the required changes, improving significantly the quality of the paper.

Reviewers' comments:

Reviewer's Responses to Questions

**Comments to the Author**

1. If the authors have adequately addressed your comments raised in a previous round of review and you feel that this manuscript is now acceptable for publication, you may indicate that here to bypass the “Comments to the Author” section, enter your conflict of interest statement in the “Confidential to Editor” section, and submit your "Accept" recommendation.

Reviewer #3: All comments have been addressed

Reviewer #4: All comments have been addressed

2. Is the manuscript technically sound, and do the data support the conclusions?

Reviewer #3: Yes

Reviewer #4: Yes

3. Has the statistical analysis been performed appropriately and rigorously? 

Reviewer #3: Yes

Reviewer #4: Yes

4. Have the authors made all data underlying the findings in their manuscript fully available?

Reviewer #3: Yes

Reviewer #4: Yes

5. Is the manuscript presented in an intelligible fashion and written in standard English?

Reviewer #3: Yes

Reviewer #4: Yes

6. Review Comments to the Author

Reviewer #3: The Authors addressed all the comments and significantly improved the quality of the manuscript.

I suggest acceptance in the present form

Regards

Reviewer #4: I read with great interest the manuscript titled "Patient's experience with Arabin cervical pessary during pregnancy: a questionnaire survey" (PONE-D-21-15296_R1).

The topic of this manuscript falls within the scope of Plos One.

I was particularly pleased to review this paper. In my honest opinion, the topic is interesting enough to attract the readers' attention. The methodology is accurate, and the data analysis supports conclusions. Moreover, authors addressed all suggested revisions, and I appreciated the manuscript improvement.

7. PLOS authors have the option to publish the peer review history of their article (what does this mean?). If published, this will include your full peer review and any attached files.

Reviewer #3: No

Reviewer #4: No

---

## [Editor Report · Acceptance letter]

22 Dec 2021

PONE-D-21-15296R1 

Patient’s experience with the Arabin cervical pessary during pregnancy: a questionnaire survey 

Dear Dr. Di Tommaso:

I'm pleased to inform you that your manuscript has been deemed suitable for publication in PLOS ONE. Congratulations! Your manuscript is now with our production department. 

Kind regards, 

on behalf of

Dr. Antonio Simone Laganà 

Academic Editor

PLOS ONE